# Automated Ground Truth Generation for Learning-Based Crack Detection on Concrete Surfaces

**Hsiang-Chieh Chen \*** and **Zheng-Ting Li**

Department of Electrical Engineering, National United University, Miaoli 360301, Taiwan; gn00920581@gmail.com
\* Correspondence: chc@nuu.edu.tw

**Abstract:** This article introduces an automated data-labeling approach for generating crack ground truths (GTs) within concrete images. The main algorithm includes generating first-round GTs, pre-training a deep learning-based model, and generating second-round GTs. On the basis of the generated second-round GTs of the training data, a learning-based crack detection model can be trained in a self-supervised manner. The pre-trained deep learning-based model is effective for crack detection after it is re-trained using the second-round GTs. The main contribution of this study is the proposal of an automated GT generation process for training a crack detection model at the pixel level. Experimental results show that the second-round GTs are similar to manually marked labels. Accordingly, the cost of implementing learning-based methods is reduced significantly because data labeling by humans is not necessitated.

**Keywords:** automated data labeling; crack detection; crack segmentation; deep learning; ground truth generation



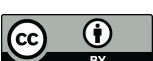

## 1. Introduction

Cracks appear on the surface of concrete structures owing to various causes, such as aging, environmental, and loading effects. Surface cracks are one of the earliest indicators of structural damage; therefore, crack monitoring has become a critical task in structural maintenance. Conventional monitoring relies on well-trained human inspectors who observe and record crack information, and hence is considered inefficient. Moreover, manual inspection results depend significantly on individual subjectivity, which may result in inaccuracies and mistakes [1]. To perform an efficient and objective crack assessment, automated inspection methods and systems must be developed. Automated crack detection can be achieved using non-destructive techniques, such as infrared thermography [2], ultrasonic sensors [3], terrestrial laser scanning [4–6], and laser displacement sensors [7].

In recent decades, image-based crack detection technology has garnered increasing interest for facilitating visual inspection on the surface of concrete. Early image-based methods were primarily devised on the basis of image processing techniques [8], such as segmentation [9], edge detection [10], filtering [11], and histogram analysis [12]. However, it is difficult to design a universal method to accommodate diverse scenes because cracks often appear in irregular patterns. In recent years, the rapid development of artificial intelligence has resulted in the extensive investigation of machine learning-based methods. Deep learning models, particularly convolution neural networks (CNNs), have demonstrated their superior performance in various computer vision applications. CNNs perform well in image classification, segmentation, and object detection tasks, as well as in image feature extraction. Inspired by the success of CNNs, some deep learning methods that split an image into patches and then employ a CNN to extract features and predict whether cracks exist within the patches have been proposed [13–15]. Although these methods can locate cracks using a patch or a bounding box, they cannot accurately identify cracks at the pixel level. Studies regarding pixel-level crack segmentation have increased significantly in recent years. Such pixel-level segmentation is often categorized into semantic

segmentation and instance segmentation. For crack detection, semantic segmentation, which performs pixel-wise identification for all target objects of the crack/non-crack class, is typically preferred.

Fully convolutional networks (FCNs) trained via end-to-end learning first demonstrated state-of-the-art performances in 2015 [16]. Subsequently, they were investigated extensively to solve challenging problems in semantic segmentation, such as crack detection at the pixel level. FCNs are an extension of the original CNN and achieve pixel-level classification on the basis of convolution layers [1]. Dung and Anh [17] proposed an FCN with an encoder, i.e., the VGG16 [18], for concrete crack detection and density evaluation. Furthermore, the crack path was accurately monitored using an FCN-based method. In [19], a deep CNN comprising an FCN with an encoder–decoder architecture was proposed for pixel-wise classification to detect cracks. Similarly, the VGG-Net was adopted in this FCN model. Zhang et al. [20] introduced an FCN based on a dilated convolution operation. They used a residual network [21] to obtain the feature maps of an input image and performed the dilated convolution operations with different dilation rates to extract feature maps under different receptive fields. CrackSegNet is an end-to-end deep network that combines multiple techniques, including feature extraction by convolution, receptive field expansion by dilated convolution, multi-scale max-pooling, and skipped connections of feature map fusion [1]. Compared with the existing FCN-based models, it offers the advantages of low generalization error and less data requirement; however, it requires more time for the inference process. On the basis of the well-known segmentation model SegNet [22], Zou et al. [23] developed DeepCrack, which is a deep CNN, by learning high-level features for crack representation. To obtain both sparse and continuous features in each scale, DeepCrack added skip layers to connect the encoder to the decoder. The effectiveness of this skipping mechanism in distinguishing cracks from the background within an image was validated.

To accommodate binary semantic segmentation, the U-Net was first proposed in [24] to achieve better segmentation results on biomedical images. The U-Net requires relatively few annotated images since it utilizes the elastic deformations of the training samples. The success of the U-Net for biomedical images motivated researchers to evaluate the performance of the U-Net in different applications, such as crack detection. Liu et al. [25] were the first to apply the U-Net to detect concrete cracks. Furthermore, they verified that the U-Net outperformed other existing deep CNNs in terms of robustness, effectiveness, and detection accuracy. Hitherto, both FCNs and U-Nets have been investigated extensively, e.g., the automatic pixel-level crack detection network [26] and the convolutional encoder–decoder network [27].

Although the methods used yielded good performance in concrete crack detection, they require a significant amount of development to be usable in practical applications. For instance, considerable effort is required in acquiring training data, particularly for annotating images in semantic segmentation. Each pixel of interest is labeled with the class of its enclosing region using annotation tools. Hence, another critical issue in crack detection segmentation is data labeling for the training set. Zou et al. [28] presented a pseudo-labeling technique to generate structured pseudo-labels with unlabeled or weakly labeled data. In [29], a self-supervised structure learning network that can be trained without using a GT was introduced. This is achieved by training a reverse network to return the output to the input. On the basis of these studies, we believe that an appropriate algorithm that can generate GTs for training data is equally important as a crack detection model that must be trained in a supervised manner. Therefore, an algorithm for generating the GTs of concrete images that can be further used for training deep learning networks to perform crack detection is proposed herein. The main contributions of this study are summarized below:

1. We introduce an algorithm that can perform automated data labeling for concrete images exhibiting cracks. This algorithm first produces preliminary labels via several

image processing procedures. Hence, the preliminary labels, namely, the first-round GTs, are used to train a deep U-Net-based model.

2.  The U-Net-based model above is implemented by integrating the VGG16 into the U-Net to form the vanilla architecture of our proposed crack detection model. In addition, the encoder portion of this crack detection model is replaced by the well-known residual network (ResNet) for evaluating the effectiveness among different encoder backbones.

3.  We propose a scheme to refine the first-round GTs to generate refined (also known as second-round) GTs. Using a fuzzy inference system and using a crack image and its prediction result yielded by the proposed model as inputs, we can derive the degree of each pixel belonging to the crack class. Next, a thresholding operation is employed to determine whether a pixel is categorized as a crack or non-crack. Subsequently, the second-round GTs of the training data were obtained. Moreover, the aforementioned U-Net-based model can be retrained using the second-round GTs to achieve better performances.

To summarize, the main contribution of this study is the proposal of an automated labeling technique that involves a three-stage procedure, including first-round GT generation, pre-training of a U-Net-based model, and second-round GT generation. The remainder of this paper is organized as follows: Section 2 introduces the main algorithm of the proposed method. In Section 3, we describe the implementation details and provide a discussion regarding the experiments. Section 4 presents the quantitative results for verifying the effectiveness of the proposed method. Finally, the conclusions are provided in the final section.

## 2. Proposed Method

This section presents a self-supervised learning approach for training a deep learning-based model for detecting cracks in concrete images. The highlight of the approach is a three-stage process for performing automated data labeling, including first-round GT generation, pre-training a U-Net-based model, and second-round GT generation. The main algorithm of the proposed method includes the following steps. For every sample in the training data, the label of cracks, namely, the first-round GT, was first generated via our automated data-labeling method. Subsequently, a deep learning-based model was pre-trained and used to detect cracks. On the basis of the fuzzy inferencing, we used a binary crack classification method for each pixel to refine the crack detection results. Finally, the refined results were considered to be second-round GTs that can be further used for re-training the crack detection model or for training any learning-based model. The entire procedure is described in detail next.

### 2.1. First-Round GT Generation

This subsection introduces an effective method for producing labels that can be used as supervisory signals to pre-train a deep learning neural network. Crack detection in an image is often regarded as a problem in binary semantic segmentation. Specifically, it is a pixel-level classification of crack and non-crack cases. Because cracks are visually presented in piecewise linear or curvilinear segments, they can be easily located by applying an edge detection algorithm. Let $I$ be the original image (size of $w \times h$ pixels), and there are $N$ images in the dataset. The main steps of our crack localization method are described as follows.

#### 2.1.1. Edge Pixel Enhancement

The original image $I$ is first converted into a grayscale image $I_g$. Subsequently, a Gaussian blur filter with a standard deviation $\sigma_G$ is applied to the grayscale image. Next, this blurred image $I_{blur}$ is subtracted from the original grayscale image $I_g$ to extract the edge points occupied, denoted by $I_e = I_g - I_{blur}$. Figure 1 shows an example of the original color images (randomly selected from the dataset provided in [30]) and the result obtained

after edge pixel enhancement. To facilitate observation, we multiplied the pixel intensity in subplot (b) by 5. It was observed that the pixels around the cracks were enhanced.

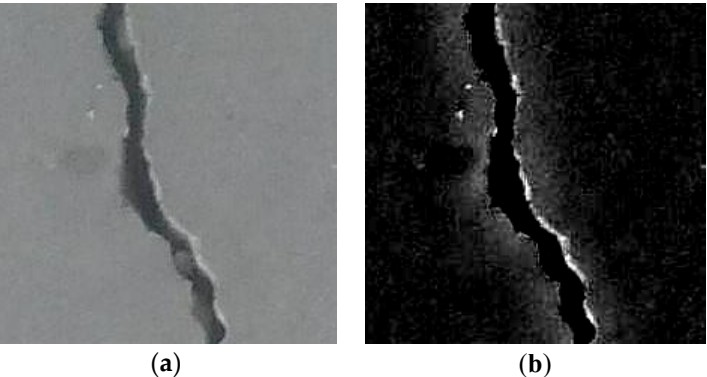

(**a**)          (**b**)

**Figure 1.** An example of edge pixel enhancement: (**a**) the original color image; (**b**) result of edge pixel enhancement.

### 2.1.2. Crack Pixel Segmentation

Intuitively, edge points often appear around cracks, and the different intensities of the grayscale represent different levels of discontinuities. Hence, a typical edge detection filter, i.e., the Sobel operator, is applied to image $I_e$. The Sobel operator uses a pair of kernels, $S_x$ and $S_y$, as presented in (1), to calculate the approximations of the derivatives in the $x$- and $y$-axes, respectively.

$$S_x = \begin{bmatrix} 1 & 0 & -1 \\ 2 & 0 & -2 \\ 1 & 0 & -1 \end{bmatrix} \text{ and } S_y = \begin{bmatrix} 1 & 2 & 1 \\ 0 & 0 & 0 \\ -1 & -2 & -1 \end{bmatrix}. \tag{1}$$

On the basis of the results of employing these two kernels convolved with $I_e$, the resulting gradients along the two axes, $G_x$ and $G_y$, are further combined to form the magnitude and direction angle, as calculated using (2) and (3).

$$Mag = \sqrt{(S_x * I_e)^2 + (S_y * I_e)^2} = \sqrt{(G_x)^2 + (G_x)^2} \tag{2}$$

$$\Theta = \tan^{-1}\left(\frac{G_y}{G_x}\right) \tag{3}$$

Subsequently, the magnitude image is obtained via a thresholding process such that the pixels whose intensity is less than a predefined threshold $T_{mag}$ become zero. The thresholding used in this study is presented in (4).

$$Mag(x,y) = \begin{cases} Mag(x,y), & \text{if } Mag(x,y) \geq T_{mag}; \\ 0, & \text{otherwise,} \end{cases} \tag{4}$$

where $Mag(x,y)$ is the intensity at pixel $(x,y)$ in the magnitude image, and $T_{mag}$ is a predefined threshold. Subsequently, the well-known morphological closing operation with a filter size of $N \times N$ is performed on this thresholded image to fill the cracks. Figure 2 shows the result of thresholding the magnitude image and the result obtained after performing the closing operation. In this study, we set the kernel size of closing operation $N = 15$, which is sufficiently large to connect the edge points to form connected components in the image. As shown in Figure 2b, the approximate crack shape can be extracted.

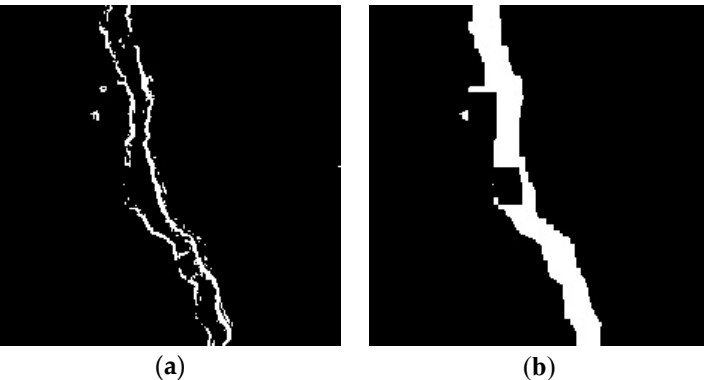

**Figure 2.** Results of (**a**) thresholding the gradient image; (**b**) performing a closing operation.

Thus far, the crack is outlined roughly. Furthermore, a more accurate label needs to be drawn to form the GT. As shown in Figure 1, the crack can be considered the foreground, whereas the flat concrete surface is the background. GrabCut [31] is an iterative image segmentation method inspired by the graph cut algorithm [32] and involves simple user interactions. The simplest method to interact with a user is by drawing a rectangle to bound the desired object. Because the crack object has been extracted, as shown in Figure 2b, the main task is to identify a rectangle to enclose the bright connected objects by excluding noise. Using the well-known connected-component labeling (CCL) method, we can locate a significant portion of the crack by a bounding box. Figure 3 presents the CCL results of Figure 2b, and the same bounding box drawn in Figure 1a. It is noteworthy that the nearby bounding boxes were merged and only the bounding boxes whose width (or height) exceeded $w/3$ (or $h/3$) were preserved.

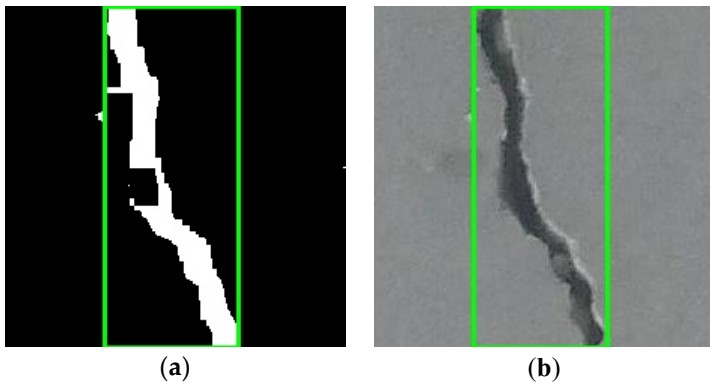

**Figure 3.** Results of connected component labeling of (**a**) Figure 2b; (**b**) a significant portion of the crack in Figure 1a.

In this study, we employed only the hard segmentation of the GrabCut algorithm because the crack boundary typically contains high-intensity discontinuities. On the basis of the bounding box shown in Figure 3b, we implemented the GrabCut algorithm within an image as follows:

Step 1: Generate an initial trimap using the bounding box.

The trimap $T = \{T_\text{B}, T_\text{F}, T_\text{U}\}$, where $T_\text{B}$, $T_\text{F}$ are background and foreground regions, respectively, and $T_\text{U}$ is the unknown region. In this initialization step, only $T_\text{B}$ was provided. The pixels outside the rectangle were marked as the background, and the foreground was set to empty, i.e., $T_\text{F} = \phi$. Hence, $T_\text{U} = \overline{T}_\text{B}$, which implies that the pixels were inside the rectangle.

Step 2: Perform an initial segmentation of the original color image.

All background pixels were categorized into the background class. Unmarked pixels were tentatively categorized into the foreground class.

Step 3: Initialize two Gaussian mixture models (GMMs).



The background and foreground GMMs, denoted by $GMM_{BG}$ and $GMM_{FG}$, respectively, were created separately using the initial segmentation results obtained in the previous step. Every GMM was a full-covariance Gaussian mixture with $K$ components. The $k$-th component comprised four parameters: the mean $\mu_k$ (a tuple in the RGB color space), covariance matrix $\Sigma$ (a $3 \times 3$ matrix), determinant of the covariance matrix $|\Sigma|$, and component weight $\varpi$.

Step 4: Assign each pixel belonging to the foreground class to the Gaussian component.

Each pixel in the foreground class was assigned to the component $GMM_{FG}$, which has the maximum likelihood. This was performed by evaluating the Gaussian distribution on the basis of considering the pixel's RGB triple as the input. Similarly, each pixel in the background class was assigned to the maximum-likelihood component of $GMM_{BG}$.

Step 5: Learn new parameters of GMMs.

The current GMMs were discarded, and the new parameters of the background and foreground GMMs were learned using the pixel sets that were assigned in Step 4.

Step 6: Conduct image segmentation.

Graph cut was performed to obtain a new tentative classification of the background and foreground for all pixels.

The entire GrabCut procedure is presented above. The iterative scheme begins with repeating Steps 4–6 until the classification result converges. Figure 4 shows the result of GrabCut from Figure 3b, in which the pixels belonging to the foreground class are preserved, and the background pixels are set to zero (black pixels). The foreground pixels are further refined by categorizing them into brighter and darker groups, and only the darker pixels are regarded as cracks. Therefore, the crack pixels in the original image are indicated. Figure 5 shows the preliminary result of crack segmentation by employing the GrabCut method and image processing techniques. It is noteworthy that the crack pixels are depicted by white pixels and can be directly applied to train a deep learning-based model since the crack and non-crack pixels are labeled by 1 and 0, respectively. In this study, we named this binary map the first-round GT.

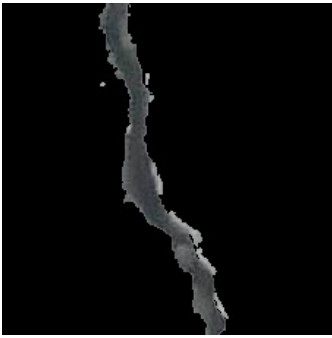

**Figure 4.** Result of foreground extraction by GrabCut algorithm from Figure 3b.

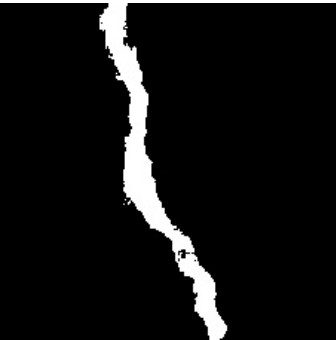

**Figure 5.** Preliminary result of crack segmentation by preserving darker pixels in the foreground from Figure 4. Crack pixels are denoted by white color in this plot.

### 2.2. Pre-Training Binary Segmentation Model for Crack Detection

Crack detection is typically achieved through binary semantic segmentation, which classifies every pixel into two classes: crack and non-crack. In this study, we first implemented a pixel-level crack detection method based on the U-Net [24], which relies on an encoder–decoder architecture and uses copy-and-crop operations to propagate the details from the encoder layers to their corresponding layers in the decoder. The crack detection model used in this study was a hybrid of U-Net and VGG16 [18], which uses VGG16 as the encoder portion of the U-Net. Additionally, the U-net encoder can be replaced by different backbones, such as the ResNet [21]. The use of different backbones will be discussed later. Figure 6 shows the architecture of the employed model presented in this subsection, and its detailed composition is presented in Table 1. Because of the limitations of the U-Net, its input size must be a multiple of 32. Hence, an input image measuring $448 \times 448$ pixels with three channels was used, and the output image was a binary map with $448 \times 448$ pixels. The symbol $\oplus$ denotes the concatenation operation. Additionally, the ReLu function was used as an activation function in every convolution layer, and an up-sampling layer was achieved by performing the well-known bilinear interpolation with a scaling factor of 2.

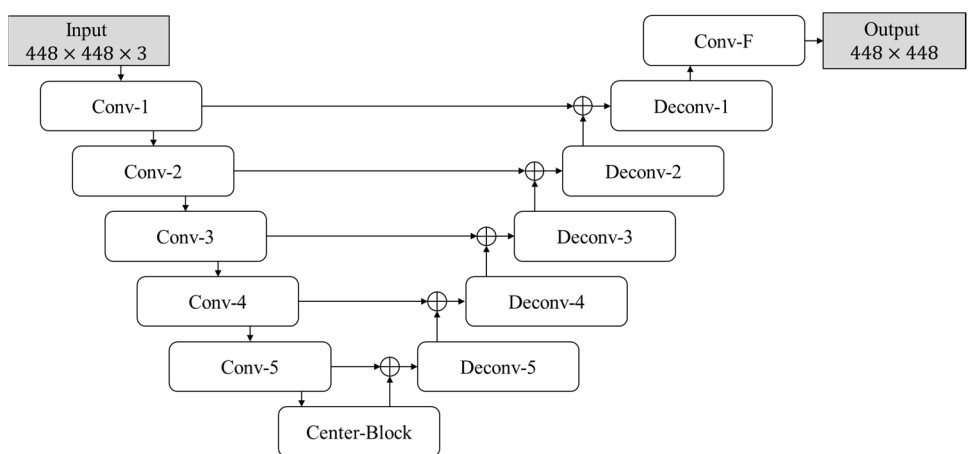

**Figure 6.** Simple architecture of the employed U-Net-based model.

Next, we used the dataset presented in [30] as the target and randomly split all of its 20,000 images into training, validation, and test sets, according to the ratio of 6:1:3. Specifically, 12,044 images were used for training, 2123 for validation, and 5833 for testing, as summarized in Table 2. To consider such a crack detection problem, we adopted the binary cross-entropy loss expressed in (5) as the loss function during training.

$$\mathcal{L}oss = -\frac{1}{N}\sum_{i=1}^{N}(z_i \log(\mathcal{S}(z_i)) + (1 - z_i)\log(1 - \mathcal{S}(z_i))) \tag{5}$$

where $N$ is the number of samples, $z_i$ is the class which is either 0 or 1, and $\mathcal{S}(\cdot)$ is the sigmoid function. The optimization used in our learning is a stochastic gradient descent with a momentum factor of 0.9, and a learning rate $\eta = 10^{-3}$. In addition, the loss function is modified by adding an L2 regularization term with a weight $\lambda = 10^{-4}$ for preventing overfitting. Figure 7 shows the per-epoch trend of the training and validation loss. It is noteworthy that the minimum total loss occurred at epoch 20, and the validation loss converged; the total loss was 0.07131. Therefore, we selected the trained model after this epoch was completed as the interim best model and applied it to detect cracks in images. The inference output of this model was a probabilistic map $I_{\mathrm{pred}}$ whose pixel value ranged from 0 to 1 and represented the probability of a pixel belonging to the crack class. To facilitate further discussions, we multiplied the pixel value of the probabilistic map by 255 and then obtained a normalized prediction map $\widetilde{I}_{\mathrm{pred}}$. Figure 8 shows the normalized prediction result of Figure 1a. Five additional representative examples are shown in

Figure 9, namely, the upper, middle, and bottom rows, which are the original images; their first-round GTs; and the detection results obtained using our pre-trained model.

**Table 1.** Complete composition of the employed U-Net-based model.

| Block Name | Layer | Kernel Size | Stride | Channels |
|---|---|---|---|---|
| Con-1 | Convolution | 3 × 3 | 1 | 3→64 |
| | Convolution | 3 × 3 | 1 | 64→64 |
| | Maxpool | 2 × 2 | 2 | - |
| Conv-2 | Convolution | 3 × 3 | 1 | 64→128 |
| | Convolution | 3 × 3 | 1 | 128→128 |
| | Maxpool | 2 × 2 | 2 | - |
| Conv-3 | Convolution | 3 × 3 | 1 | 128→256 |
| | Convolution | 3 × 3 | 1 | 256→256 |
| | Convolution | 3 × 3 | 1 | 256→256 |
| | Maxpool | 2 × 2 | 2 | - |
| Conv-4 | Convolution | 3 × 3 | 1 | 256→512 |
| | Convolution | 3 × 3 | 1 | 512→512 |
| | Convolution | 3 × 3 | 1 | 512→512 |
| | Maxpool | 2 × 2 | 2 | - |
| Conv-5 | Convolution | 3 × 3 | 1 | 512→512 |
| | Convolution | 3 × 3 | 1 | 512→512 |
| | Convolution | 3 × 3 | 1 | 512→512 |
| | Maxpool | 2 × 2 | 2 | - |
| Center-Block | Up-sampling | 2 × 2 | Scale factor: 2 | - |
| | Convolution | 3 × 3 | 1 | 512→512 |
| | Convolution | 3 × 3 | 1 | 512→256 |
| Deconv-5 | Up-sampling | 2 × 2 | Scale factor: 2 | - |
| | Convolution | 3 × 3 | 1 | 768→512 |
| | Convolution | 3 × 3 | 1 | 512→256 |
| Deconv-4 | Up-sampling | 2 × 2 | Scale factor: 2 | - |
| | Convolution | 3 × 3 | 1 | 7698→512 |
| | Convolution | 3 × 3 | 1 | 512→256 |
| Dconv-3 | Up-sampling | 2 × 2 | Scale factor: 2 | - |
| | Convolution | 3 × 3 | 1 | 512→256 |
| | Convolution | 3 × 3 | 1 | 256→64 |
| Deconv-2 | Up-sampling | 2 × 2 | Scale factor: 2 | - |
| | Convolution | 3 × 3 | 1 | 192→128 |
| | Convolution | 3 × 3 | 1 | 128→32 |
| Deconv-1 | Convolution | 3 × 3 | 1 | 96→32 |
| Conv-F | Convolution | 3 × 3 | 1 | 32→1 |

**Table 2.** Data distribution in the dataset.

| Category | Ratio | Number of Samples | Percentage |
|---|---|---|---|
| Training | 6 | 12,044 | 60.22% |
| Validation | 1 | 2123 | 10.615% |
| Test | 3 | 5833 | 29.165% |
| Total | 10 | 20,000 | 100% |

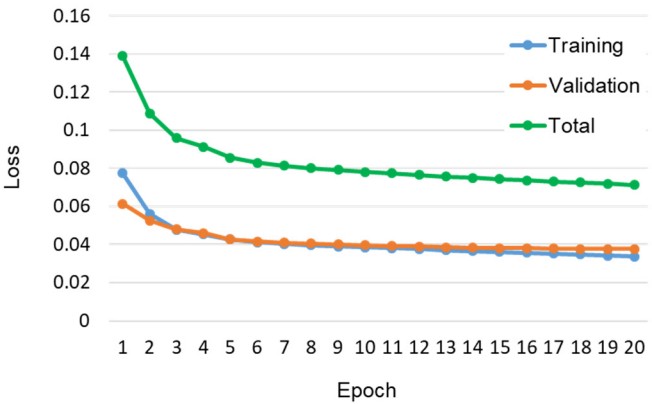

**Figure 7.** Trend of training and validation loss.

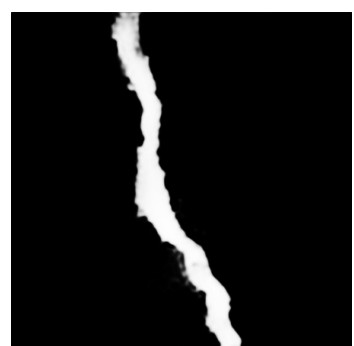

**Figure 8.** Crack detection result from Figure 1a, obtained using our proposed model.

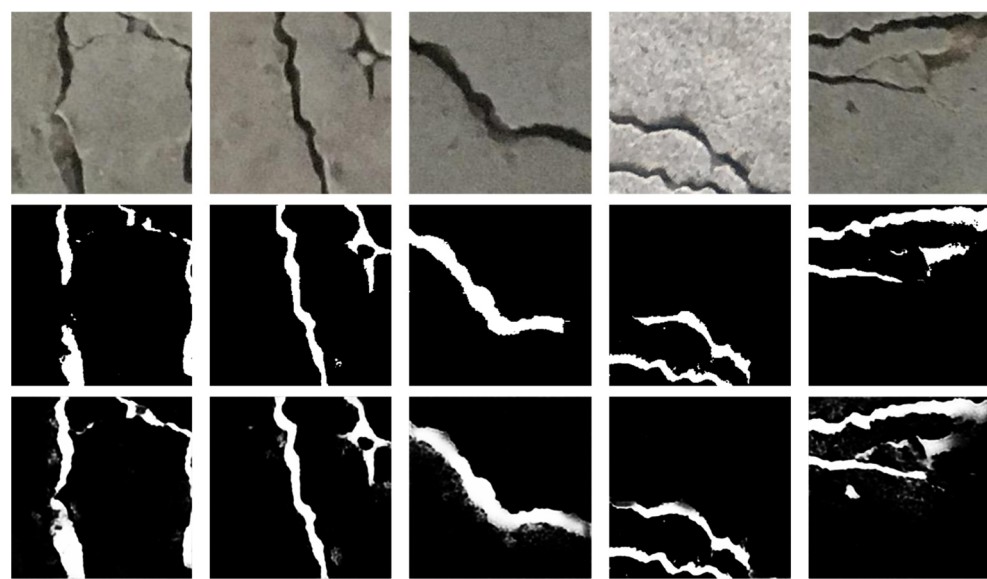

**Figure 9.** Five additional examples for representing the performance of our pre-trained model.

### 2.3. Second-Round GT Generation: Refinement Stage

As shown in Figure 9, the performance of the pre-trained model appeared to be acceptable for conducting crack segmentation, even though the automated labeled data were not comparable to the manually labeled data. The next stage is to obtain more accurate GTs for re-training the model. The entire procedure for achieving this goal is described as follows:

First, the grayscale image $I_g$ was enhanced through contrast-limited adaptive histogram equalization (CLAHE) [33], and the enhanced image is denoted as $\widetilde{I}_g$. CLAHE is

a widely used method for contrast enhancement and has been verified to be effective in several applications [34,35]. In this method, an image is divided into non-overlapping regions (also known as tiles) of equal size, and the histogram equalization per tile is operated separately. Next, a simple bilinear interpolation was employed to eliminate the inconsistent boundaries between the tiles. In this study, we determined the hyper-parameters for CLAHE, i.e., a clip limit of 0.1 and a tile size of $8 \times 8$ pixels, on the basis of several experimental tests.

Second, a fuzzy inference system (FIS) is proposed to determine whether a specified pixel $(x, y)$ is labeled as a crack in the second-round GTs. Let $p_1 = \widetilde{I}_g(x, y)$ and $p_2 = \widetilde{I}_{\text{pred}}(x, y)$ be two antecedent variables of the proposed FIS, and $q$ be its consequent variable. Here, $p_1$ represents the intensity of pixel $(x, y)$ in the enhanced grayscale image $\widetilde{I}_g$, and $p_2$ is the pixel value of the normalized map $\widetilde{I}_{\text{Pred}}$ at the same position. Both antecedent variables range from 0 to 255, and their fuzzy sets are depicted in Figure 10, in which the trapezoidal and triangular functions are used as the membership functions. For the consequent part, $q$ is represented by equally spaced triangular membership functions, as plotted in Figure 11. The linguistic terms include *Very Small* (VS), *Small* (S), *Medium* (M), *Large* (L), and *Very Large* (VL). The parameters for defining the five membership functions of the two antecedent variables are $\left\{ \alpha_1^k \mid k = 1, 2, \ldots, 5 \right\}$ and $\left\{ \alpha_2^k \mid k = 1, 2, \ldots, 5 \right\}$. For simplicity, we only attempted to obtain the values of $\alpha_1^5$ and $\alpha_2^5$, and set $\alpha_1^1 = \alpha_2^1 = 0$; meanwhile, the other parameters were equally spaced between them. Table 3 lists the parameters determined using the trial-and-error analysis method used in this work.

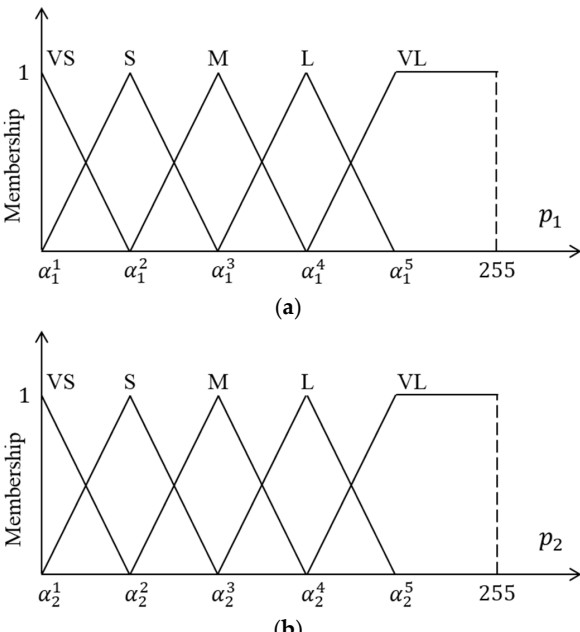

**Figure 10.** Fuzzy sets of antecedent variables.

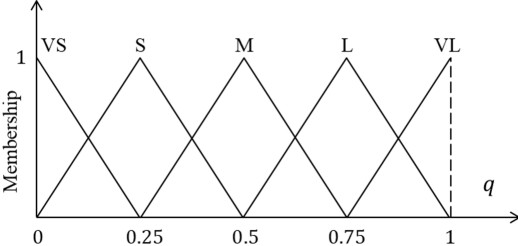

**Figure 11.** Fuzzy sets of consequent variables.

**Table 3.** Parameters of fuzzy sets of antecedent variables.

| | $\alpha_1^1$ | $\alpha_1^2$ | $\alpha_1^3$ | $\alpha_1^4$ | $\alpha_1^5$ |
|---|---|---|---|---|---|
| Variable $p_1$ | 0 | 40 | 80 | 120 | 160 |
| | $\alpha_2^1$ | $\alpha_2^2$ | $\alpha_2^3$ | $\alpha_2^4$ | $\alpha_2^5$ |
| Variable $p_2$ | 0 | 50 | 100 | 150 | 200 |

According to the physical characteristics of cracks within an image, the pixels belonging to cracks are often presented in darker colors compared with their neighbors, which belong to the non-crack class. On the basis of the description above and the normalized prediction map $\widetilde{I}_{\mathrm{Pred}}$, we used variables $p_1$ and $p_2$ as two inputs and objectively constructed fuzzy rules using the aforementioned linguistic terms. For example:

**IF** $p_1$ is *Small* (S) **AND** $p_2$ is *Large* (L), **THEN** $q$ is *Large* (L).

Here, the consequent variable $q$ indicate the degree to which pixel $(x, y)$ is considered to be the crack class. The entire fuzzy rule base is tabulated in Table 4, in which 25 rules are included. The $m$-th fuzzy rule can be formally written in a canonical format as follows:

***Rule** m*: **IF** $p_1$ is $\hat{A}_1^m$ **AND** $p_2$ is $\hat{A}_2^m$, **THEN** $q$ is $\hat{B}^m$.

**Table 4.** Fuzzy rule table for determining the degree to which a pixel belongs to the crack class.

| $p_2$ \ $p_1$ | VS | S | M | L | VL |
|---|---|---|---|---|---|
| VS | M | VS | VS | VS | VS |
| S | M | S | S | VS | VS |
| M | L | M | S | S | VS |
| L | L | L | M | S | VS |
| VL | VL | L | M | M | S |

Here, $m = 1, 2, \ldots, 25$, and $\hat{A}_1^m \in \{\mathrm{VS}, \mathrm{S}, \mathrm{M}, \mathrm{L}, \mathrm{VL}\}$, $\hat{A}_1^m \in \{\mathrm{VS}, \mathrm{S}, \mathrm{M}, \mathrm{L}, \mathrm{VL}\}$, and $\hat{B}^m \in \{\mathrm{VS}, \mathrm{S}, \mathrm{M}, \mathrm{L}, \mathrm{VL}\}$ are selected from the fuzzy sets of $p_1$, $p_2$, and $q$, respectively. Whereas an input pair $(\widetilde{p}_1, \widetilde{p}_2)$ is imported into the FIS and fires some of the fuzzy rules, the non-fuzzy output can be obtained using the minimum inference engine and the center-of-gravity defuzzification method [36], as formulated below.

$$\widetilde{q} = \frac{\int_Q q B'(q) dq}{\int_Q B'(q) dq} \tag{6}$$

and

$$B'(q) = \max_m \{A_1^m(\widetilde{p}_1) \wedge A_2^m(\widetilde{p}_2) \wedge G^m(q)\}, \tag{7}$$

where $\wedge$ is the minimum operator, and Q is the universe of discourse.

Therefore, the main procedure for generating a refined GT is to employ our proposed FIS to classify each pixel as crack or non-crack, followed by forming a binary map to be used as the GT. This is achieved as follows:

- Step 1: For a specified pixel $(\widetilde{x}, \widetilde{y})$, two inputs $\widetilde{p}_1 = \widetilde{I}_g(\widetilde{x}, \widetilde{y})$ and $\widetilde{p}_2 = \widetilde{I}_{\mathrm{pred}}(\widetilde{x}, \widetilde{y})$ are imported into the proposed FIS.
- Step 2: The non-fuzzy output $\widetilde{q}$ is obtained using the fuzzy inference engine. This output is regarded as the degree to which pixel $(\widetilde{x}, \widetilde{y})$ belongs to the crack or non-crack class.
- Step 3: Label the refined (second-round) GT, as expressed by

$$I_2^{\mathrm{GT}}(\widetilde{x}, \widetilde{y}) = \begin{cases} 1, & \text{if } \widetilde{q} \geq T_{\mathrm{crack}}; \\ 0, & \text{otherwise.} \end{cases} \tag{8}$$

The steps above are repeated for all pixels in the image, where $1 \leq \tilde{x} \leq w$ and $1 \leq \tilde{y} \leq h$. Therefore, a binary map (size of $w \times h$ pixels) can be obtained, in which the crack and non-crack pixels are denoted by 1 and 0, respectively. This map is regarded as the second-round GT and is further used for re-training the pre-trained crack detection model. To facilitate observation, Figure 12 shows the original image, as well as the first- and second-round GTs in subplots (a), (b), and (c). As shown, the shape of the second-round GT was smoother than that of the first-round GT and resembled labeling by a human. Figure 13 shows another five examples that were randomly selected from the dataset. The upper, middle, and bottom rows represent the original, first-round, and second-round GT labels, respectively.

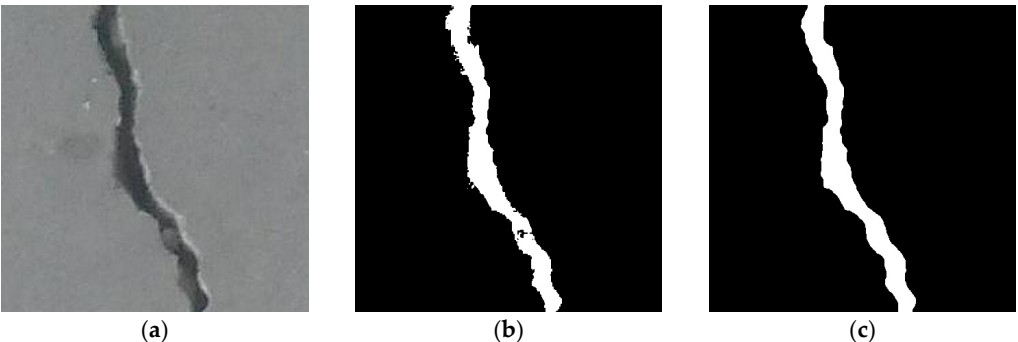

| (a) | (b) | (c) |

**Figure 12.** Example of an image with crack: (**a**) original image; (**b**) first-round crack GT; (**c**) second-round crack GT.

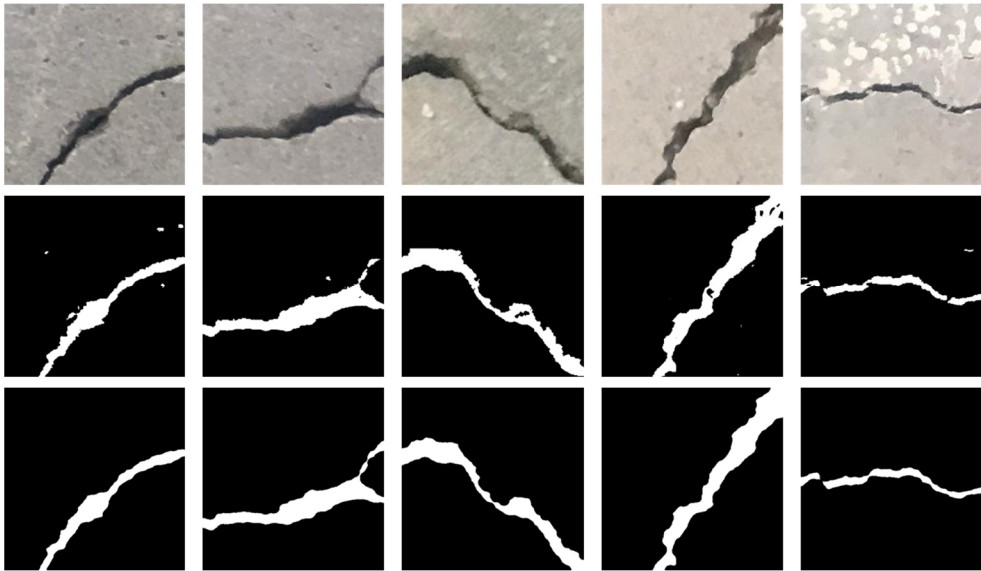

**Figure 13.** Five randomly selected examples: original image (**upper**), and their first-round GT (**middle**), and second-round GT (**bottom**).

### 2.4. Main Procedure of Proposed Algorithm

The goal of the proposed algorithm is to obtain labeled data that can be regarded as the GT for training a learning-based crack segmentation. To verify the effectiveness of our automated labeling algorithm, we implemented a deep learning model that is a hybrid of the U-Net and VGG16 to identify cracks by pixel. The configuration of the proposed algorithm is outlined, and the overall procedure for acquiring second-round GTs for a dataset is summarized as the following Algorithm 1. The implementation details and experiments are discussed in the following sections.

---

**Algorithm 1:** Automated Data Labeling for a Dataset

---

Input: All images in the dataset. Let *I* be a specific image.
Output: Second-round GTs for all images.
Steps:

   1: Convert *I* into a grayscale image $I_g$.
   2: Apply Gaussian blur filter on $I_g$, and obtain a blurred image $I_{blur}$.
   3: Subtract the blurred image $I_{blur}$ from the gray image $I_g$,
       denoted by $I_e = I_g - I_{blur}$.
   4: Perform Sobel edge detector on $I_e$, and obtain the gradient magnitude *Mag* and direction $\Theta$.
   5: Binarize the magnitude map *Mag* by thresholding.
   6: Perform closing operation on this binarized map.
   7: Use connected-component labeling to obtain bounding boxes of cracks.
   8: Apply GrabCut to extract crack pixels which are denoted by 1 in the first-round GT.
   9: Repeat Steps 1–8 for every image in the dataset. Collect training data, in which each sample consists of a pair of an image and
       its first-round GT.
  10: Pre-train a binary segmentation model using the training data obtained in Step 9.
  11: Obtain the prediction result $I_{pred}$ for the image *I* using this pre-trained model.
  12: Normalize $I_{pred}$ to $\widetilde{I}_{pred}$, in which every pixel value ranges from 0 to 255.
  13: Enhance the grayscale image $I_g$ to be $\widetilde{I}_g$ by CLAHE.
  14: For every pixel $(x, y)$ in the image *I*:Perform the proposed FIS to determine the degree to which pixel $(x, y)$ be longs to the
       crack or non-crack class.
  15: Repeat Steps 11–14 for every image in the dataset. The second-round GTs of all training samples are obtained.

---

## 3. Implementation and Experiments

The proposed algorithm was implemented on a GPU-accelerated computer with an Intel CoreTM i7-11800 @ 2.3 GHz and 32G RAM, and an NVIDIA GeForce GTX 3080 with an 8G GPU. In this section, the detailed implementation of our proposed method and the reduced computation afforded by the proposed FIS are discussed.

### 3.1. Crack Detection Models Based on U-Net

In the present study, a U-Net-based model was implemented because it is superior to other conventional methods, such as CrackTree [37], CrackIt [38], and CrackForest [39]. In Section 2.2, a hybrid architecture of the U-Net and VGG16 was introduced to perform per-pixel crack segmentation. It is noteworthy that the U-Net encoder can be replaced by different backbones. Hence, we used the ResNet [21] for the encoder portion of the U-Net (the left half in Figure 6, including the blocks named Conv-1 to Conv-5). Table 5 summarizes the complete compositions of the encoder replaced by ResNet-18, 34, 50, and 101. Therefore, the vanilla version was compared with four U-Net-based models that involve different ResNets in this study. We named them Res-U-Net-18, Res-U-Net-34, Res-U-Net-50, and Res-U-Net-101.

To evaluate the performance of these five models, we used the dataset introduced in Table 2 to train each model. Before implementing our proposed algorithm, all the images were normalized to a size of 448 × 448 pixels in advance because the width and height of the input images must be a multiple of 32 (the limitation of using the U-Net-based model). The main procedure of automated data labeling for obtaining the second-round GT is described below:

1. Perform the algorithm of the first-round GT generation proposed in Section 2.1.
2. Pre-train the U-Net-based models, including the vanilla, Res-U-Net-18, Res-U-Net-34, Res-U-Net-50, and Res-U-Net-101 models, separately. The hyper-parameters used during this training stage are the same as those introduced in Section 2.2.
3. Use each learned model to obtain the crack prediction results of the training data.
4. On the basis of the prediction results, obtain the second-round GTs using the refinement scheme presented in Section 2.3.
5. Finally, use the second-round GTs to re-train the five pre-trained models separately. Hence, U-Net-based crack detection models with different types of encoders are obtained.

**Table 5.** The complete composition of the U-Net-based models with different encoders.

| Block Names | Encoder Backbones | | | |
|---|---|---|---|---|
| | **ResNet-18** | **ResNet-34** | **ResNet-50** | **ResNet-101** |
| Conv-1 | $7 \times 7.64$, stride 2 $3 \times 3$ Maxpool, stride 2 | | | |
| Conv-2 | $\begin{bmatrix} 3 \times 3.64 \\ 3 \times 3.64 \end{bmatrix} \times 2$ | $\begin{bmatrix} 3 \times 3.64 \\ 3 \times 3.64 \end{bmatrix} \times 3$ | $\begin{bmatrix} 1 \times 1.64 \\ 3 \times 3.64 \\ 1 \times 1.64 \end{bmatrix} \times 3$ | $\begin{bmatrix} 1 \times 1.64 \\ 3 \times 3.64 \\ 1 \times 1.64 \end{bmatrix} \times 3$ |
| Conv-3 | $\begin{bmatrix} 3 \times 3.128 \\ 3 \times 3.128 \end{bmatrix} \times 2$ | $\begin{bmatrix} 3 \times 3.128 \\ 3 \times 3.128 \end{bmatrix} \times 4$ | $\begin{bmatrix} 1 \times 1.128 \\ 3 \times 3.128 \\ 1 \times 1.512 \end{bmatrix} \times 4$ | $\begin{bmatrix} 1 \times 1.128 \\ 3 \times 3.128 \\ 1 \times 1.512 \end{bmatrix} \times 4$ |
| Conv-4 | $\begin{bmatrix} 3 \times 3.256 \\ 3 \times 3.256 \end{bmatrix} \times 2$ | $\begin{bmatrix} 3 \times 3.256 \\ 3 \times 3.256 \end{bmatrix} \times 6$ | $\begin{bmatrix} 1 \times 1.256 \\ 3 \times 3.256 \\ 1 \times 1.1024 \end{bmatrix} \times 6$ | $\begin{bmatrix} 1 \times 1.256 \\ 3 \times 3.256 \\ 1 \times 1.1024 \end{bmatrix} \times 23$ |
| Conv-5 | $\begin{bmatrix} 3 \times 3.512 \\ 3 \times 3.512 \end{bmatrix} \times 2$ | $\begin{bmatrix} 3 \times 3.512 \\ 3 \times 3.512 \end{bmatrix} \times 3$ | $\begin{bmatrix} 1 \times 1.512 \\ 3 \times 3.512 \\ 1 \times 1.2048 \end{bmatrix} \times 3$ | $\begin{bmatrix} 1 \times 1.512 \\ 3 \times 3.512 \\ 1 \times 1.2048 \end{bmatrix} \times 3$ |

For analyzing the time consumption of each model, we first investigated the number of model parameters and the computation time per image (sized of $448 \times 448$ pixels). As listed in Table 6, the Res-U-Net-18 model had the fastest execution time and the smallest number of parameters, whereas the vanilla architecture's execution time was at mid-level. Figure 14 shows the per-epoch trend of training and validation losses, trained by the second-round GTs. According to a comparison of the training loss, the Res-U-Net-101 model outperformed the others because it exhibited the most complicated architecture; however, it performed the worst in terms of validation loss. This is because the image features were memorized rather than learned in that model, i.e., overfitting occurred. As shown by the plots, the vanilla structure is suitable for crack detection in our study. Figure 15 shows several examples of the proposed models, in which the first and second columns show the original images and their first-round GTs, respectively, whereas those shown from the third to the last columns are the second-round GTs for the vanilla, Res-U-Net-18, Res-U-Net-34, Res-U-Net-50, and Res-U-Net-101 models, respectively. As shown by the results of the second-round GTs, the vanilla and Res-U-Net-34 yielded similar results. The vanilla U-Net-based model exhibited good performance in crack detection problems.

**Table 6.** The number of parameters and computation time among different encoder backbones.

| Models | Backbones | Time (Unit: ms) | | | Number of Model Parameters |
|---|---|---|---|---|---|
| | | **Min.** | **Max.** | **Avg.** | |
| Vanilla | VGG16 | 49.5 | 50.4 | 49.7 | 29,306,465 |
| Res-U-Net-18 | ResNet-18 | 41.9 | 42.8 | 42.2 | 25,009,737 |
| Res-U-Net-34 | ResNet-34 | 44.4 | 44.8 | 44.5 | 35,117,897 |
| Res-U-Net-50 | ResNet-50 | 56.1 | 56.7 | 56.4 | 57,677,897 |
| Res-U-Net-101 | ResNet-101 | 63.9 | 64.5 | 64.1 | 76,670,025 |

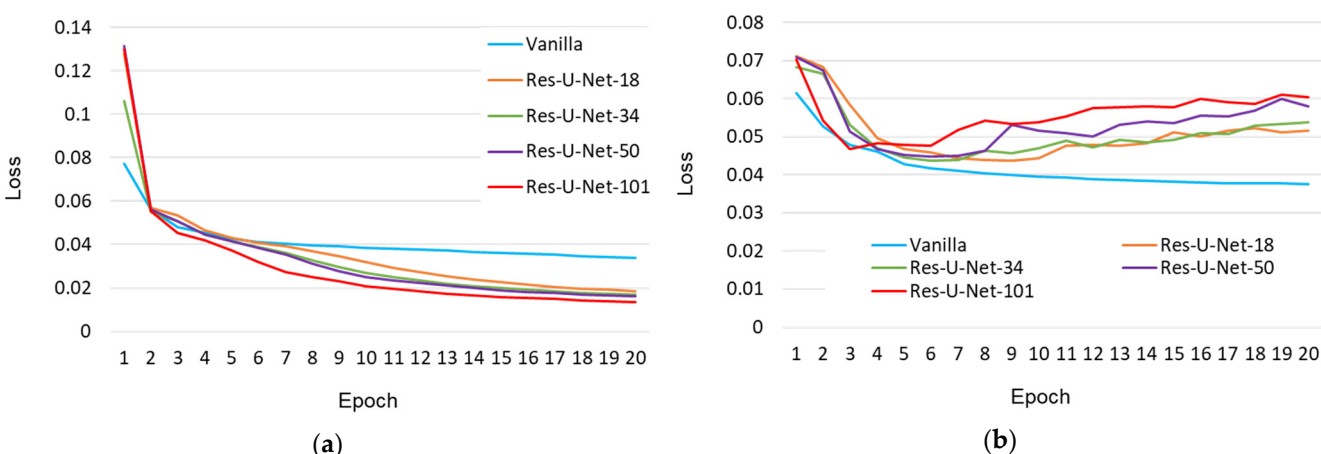

**Figure 14.** Per-epoch trend of losses: (**a**) training loss; (**b**) validation loss.

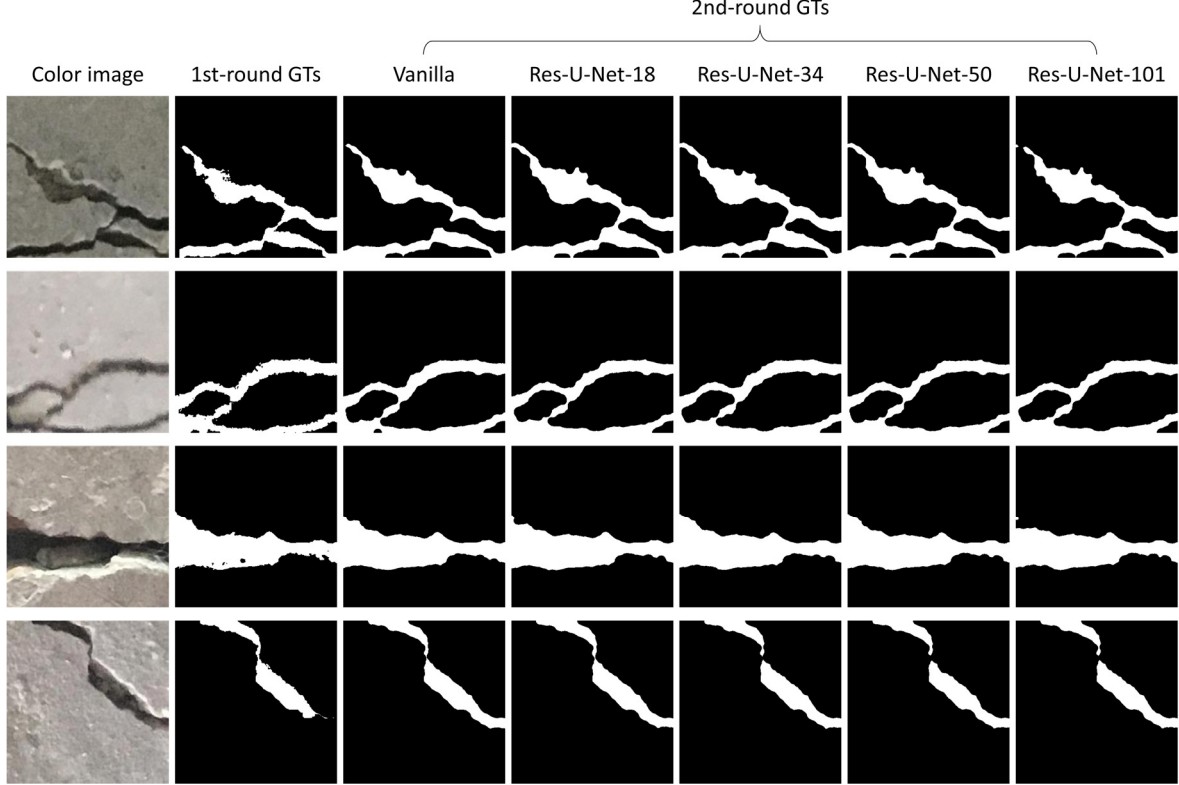

**Figure 15.** Second-round GTs obtained using five different U-Net-based models.

### 3.2. Further Discussion on Computation of FIS

In Section 2.3, an FIS was introduced to derive the degree of an interesting pixel belonging to the crack class. As an input pair goes into the FIS, several steps, including fuzzification, firing rules, inferencing, and defuzzification, are required to compute the final output. Because the process above must be performed at the pixel level, a considerable amount of computation time is required. To reduce the computation time, we transformed the proposed FIS into an input–output mapping, i.e., $\phi = \text{func}(p_1, p_2)$. This mapping can be pre-determined and constructed using a lookup table for $p_1 = 0, 1, 2, \ldots, 255$ and $p_2 = 0, 1, 2, \ldots, 255$. Figure 16 shows the pre-computed mapping surface, in which the horizontal plane is the $p_1$ vs. $p_2$ plane, and the vertical axis is the crisp output of $q$. Each pixel in the original image is classified as the crack class if its output derived from the

mapping is greater than a predefined threshold $T_{\text{crack}} = 0.4$. Accordingly, the computation time for our proposed FIS is reduced significantly by replacing the inference process with such a lookup table.

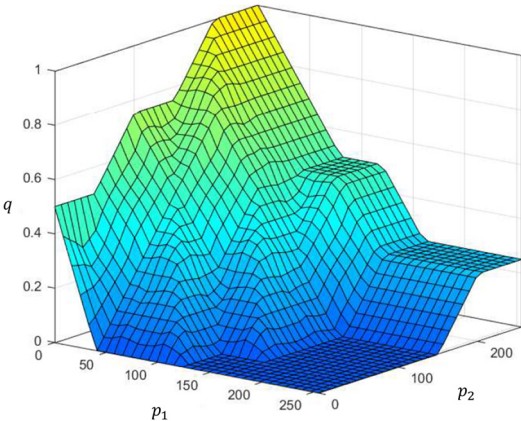

**Figure 16.** The input to output mapping surface of the proposed FIS to derive the degree to which a pixel belongs to the crack class.

## 4. Quantitative Evaluation Using Different Datasets

We generated an evaluation dataset in which the samples were collected from the concrete images of the DeepCrack [40] and edge-based labeled crack image (ELCI) [41] datasets. Because the GTs of the crack images in these two datasets were labeled by their provider, they are suitable for computing the performance metrics accurately. The intersection over union (IoU) metric was adopted as the main indicator for quantifying the performance of our proposed method. We measured the percent overlap between the annotated GT and the second-round GT yielded by our method, as expressed in (9).

$$\text{IoU} = \frac{(\text{Annotated GT}) \cap (\text{prediction result})}{(\text{Annotated GT}) \cup (\text{prediction result})} \tag{9}$$

For crack segmentation tasks, the IoU can be calculated on the basis of the true positive (TP), false positive (FP), and false negative (FN) values at the pixel level for the crack class.

$$\text{IoU} = \frac{\text{TP}}{\text{TP} + \text{FP} + \text{FN}} \tag{10}$$

The second-round GTs obtained by our proposed method are regarded as the prediction result in calculating the IoU values. In addition to the IoU value, the precision, recall, and F1-score are also computed as follows:

$$\text{Precision} = \frac{\text{TP}}{\text{TP} + \text{FP}} \tag{11}$$

$$\text{Recall} = \frac{\text{TP}}{\text{TP} + \text{FN}} \tag{12}$$

$$\text{F1} - \text{score} = \frac{2 \times \text{Precision} \times \text{Recall}}{\text{Precision} + \text{Recall}} \tag{13}$$

In the quantitative evaluation experiments, we acquired 200 concrete images and their GTs to form the evaluation dataset from DeepCrack and ELCI. Each image and its crack GT were pre-processed and normalized into $448 \times 448$ pixels. Subsequently, the proposed method was applied to obtain the second-round GTs for the concrete images of the evaluation dataset; hence, the performance metrics for 200 images were computed and averaged. As shown in Figures 14 and 15, the vanilla version of our proposed method performed well and was the least affected by overfitting; hence, it was selected as the

decisive model for computing the performance indicators. Table 7 lists the results in terms of the IoU, precision, recall, and F1-score. Because the cracks were extremely irregular and the GT was labeled manually, a small tolerance margin between the annotated GT and the prediction result can be used to measure the coincidence between the detected cracks and the GT [42]. In Table 7, the margin of *n* pixels (*n* = 1, 2, 3) was used, i.e., TP pixels were included within an *n*-pixel vicinity of the GT. The notation 0-pixel denotes that the tolerance margin is not utilized, whereas 1-pixel and 2-pixel indicate that the tolerance margins with 1 and 2 pixels were employed, respectively. As shown in Table 7, our proposed method with a vanilla architecture can achieve 94.4% precision when the tolerance margin is 2-pixel in the vicinity. Figure 17 shows five samples from the evaluation dataset, in which the upper, middle, and bottom rows represent the concrete images, the GTs labeled by humans, and the prediction results (second-round GTs obtained using our method), respectively.

**Table 7.** Numerical results obtained using vanilla version of our proposed method.

| Metrics Vicnity | IoU | Precision | Recall | F1-Score |
|---|---|---|---|---|
| 0-pixel | 0.667 | 0.723 | 0.794 | 0.778 |
| 1-pixel | 0.801 | 0.895 | 0.856 | 0.890 |
| 2-pixel | 0.814 | 0.944 | 0.883 | 0.898 |

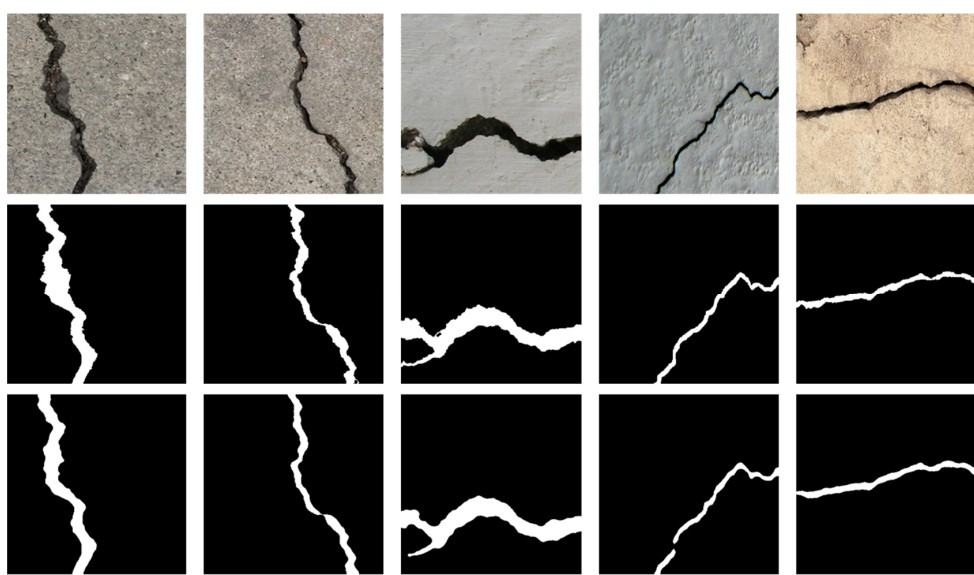

**Figure 17.** Five examples in the evaluation dataset: original image (**upper**), manually labeled GTs (**middle**), and prediction results (second-round GTs) obtained using our method (**bottom**).

## 5. Further Discussions and Improvements

As shown in Figure 17, there were minor defects that existed in the second-round GTs, i.e., the thin crack was not marked near the edge of the second-round GT (in the third column), and the crack broke into two piecewise objects (in the fourth column). The cause of these two cases is the threshold value $T_{crack}$ in (8). In the experiments above, we simply set $T_{crack} = 0.4$, which was an approximate value obtained using the trial-and-error method through evaluating all training data. A fixed threshold could not adapt to various conditions. In addition, this threshold was determined per image using the well-known Otsu's method [43] in this section. Figure 18 shows the results of the second-round GTs using different values of $T_{crack}$. The upper, middle, and bottom rows represent the prediction results of using $T_{Otsu}$, $0.9 \cdot T_{Otsu}$, and $0.7 \cdot T_{Otsu}$, respectively, for the thresholding values, where $T_{Otsu}$ was the threshold obtained by Otsu's method. It was observed that the threshold value of $0.7 \cdot T_{Otsu}$ was suitable for every image.

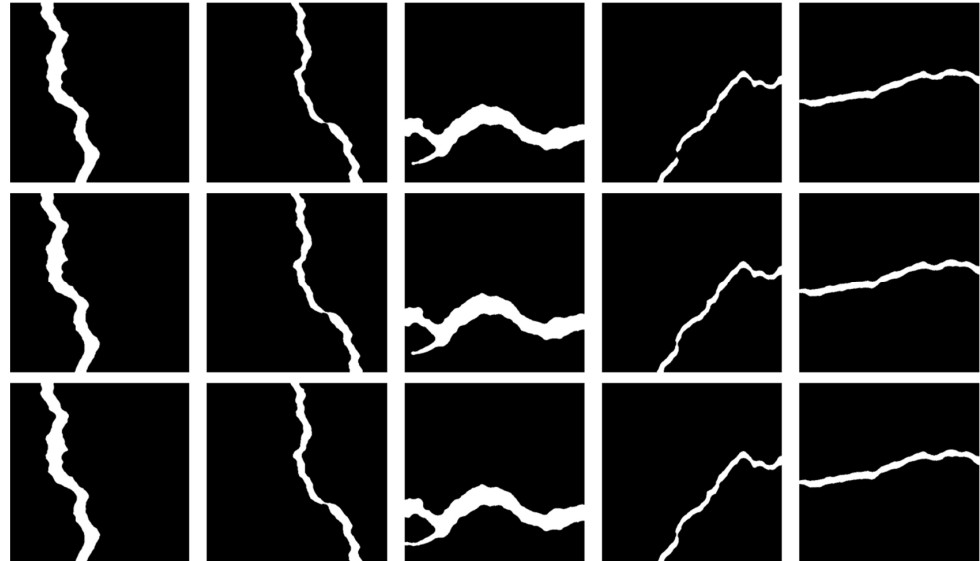

**Figure 18.** Results of second-round GTs using different values of $T_{crack}$: by Otsu's method $T_{crack} = T_{Otsu}$ (**upper**), $T_{crack} = 0.9 \cdot T_{Otsu}$ (**middle**), and $T_{crack} = 0.7 \cdot T_{Otsu}$ (**bottom**).

## 6. Conclusions

An algorithm for performing automated data labeling for concrete images with cracks is presented herein. The main procedure of the proposed algorithm included the following: (1) first-round GT generation, (2) training of a deep U-Net-based model, and (3) second-round GT generation. The refined GTs can be used to train a final model for detecting cracks on concrete surfaces. Our proposed algorithm enables the self-supervised learning of training a deep learning-based crack detection method for concrete images because the cracks can be automatically labeled at the pixel level. The experimental results showed that the second-round GTs yielded by the proposed algorithm were similar to manually labeled GTs. Therefore, any learning-based model for concrete crack detection can be trained in a self-supervised manner using the proposed method to generate GTs for training samples. Furthermore, the cost of learning will be reduced significantly as the GTs need not be labeled manually.

**Author Contributions:** Conceptualization, H.-C.C.; methodology, H.-C.C.; software, Z.-T.L.; validation, H.-C.C. and Z.-T.L.; formal analysis, H.-C.C.; investigation, Z.-T.L.; resources, H.-C.C.; writing—original draft preparation, Z.-T.L.; writing—review and editing, H.-C.C.; visualization, Z.-T.L.; supervision, H.-C.C.; project administration, H.-C.C.; funding acquisition, H.-C.C. All authors have read and agreed to the published version of the manuscript.

**Funding:** This research was funded by the Ministry of Science and Technology, Taiwan, grant numbers MOST 108-2221-E-239-026 and 110-2221-E-239-033.

**Institutional Review Board Statement:** Not applicable.

**Informed Consent Statement:** Not applicable.

**Data Availability Statement:** The data used to support the findings of this study are available from the corresponding author upon request.

**Conflicts of Interest:** The authors declare no conflict of interest.

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
