# Peer review of "Automated Ground Truth Generation for Learning-Based Crack Detection on Concrete Surfaces"

_applsci, doi:10.3390/app112210966_

Round 1
Reviewer 1 Report
The paper is well structured. The state of the art is well described and documented. The bibliography is complete.
The description of the results shows some aspects that should be better explored. In particular, figure 17 shows a crack (third column) which ends before the edge of the image, and a crack(fourth column) which is broken in two parts..
This aspect should be investigated (is it linked to the closing operation?).
Author Response
Comment 1: In particular, figure 17 shows a crack (third column) which ends before the edge of the image, and a crack(fourth column) which is broken in two parts..
This aspect should be investigated (is it linked to the closing operation?).
Answer: Thank you for pointing this out. We did not notice this defect because the crack was detected by our proposed U-Net model, but filtered out by the fuzzy inference system. The root cause was that a fixed value of threshold T_crack was used in all images. In this revised version, we added Section 5 for describing the cause and our suggestion to overcome the problem. A well-known adaptive threshold selection method (named after Otsu) was used for obtaining the threshold. Please refer to Section 5.
Reviewer 2 Report
This study presents a ground truth generation method for labeling cracks. The produced results seem very close to that human labeled. The GTs would be useful to machine-learning based methods for crack detection tasks in concrete images. However, there are several parts missed for improving this manuscript. Please consider the following suggestions to revise the paper.
- In Section 3.1, five different U-Net based methods were compared based on the training and validation loss. In addition to the loss trend to evaluate performance, the computation time is also important. Please add a part to discuss the time for inferencing a model.
- As observed from Fig. 17, it is a pity that the second-round GT broke in some cases. Can it be improved using some post processing techniques. If the GTs could be modified better, the concrete dataset produced by the proposed method will be useful, and becomes a reference when training or designing a deep-learning network.
- Before resubmitting the paper, please do a proofreading carefully.
Author Response
Comment 1: In Section 3.1, five different U-Net based methods were compared based on the training and validation loss. In addition to the loss trend to evaluate performance, the computation time is also important. Please add a part to discuss the time for inferencing a model.
Answer 1: Thank you for your suggestion. We totally agree this point. In the revised version, we added several sentences in Subsection 3.1, lines 406-410 in page 14, for describing the execution time and the summarized comparison was tabulated in new Table 7. Please refer the revised parts.
Comment 2: As observed from Fig. 17, it is a pity that the second-round GT broke in some cases. Can it be improved using some post processing techniques. If the GTs could be modified better, the concrete dataset produced by the proposed method will be useful, and becomes a reference when training or designing a deep-learning network.
Answer 2: Thank you for this suggestion. Actually, after we performed several experiments for finding the cause of this problem, the determination of the threshold value of T_crack in eq. (8) affected strongly to the final output. Therefore, we added Section 5 in the revision for introducing a threshold selection method that can be performed per image, and thus the output could be better than before. Please refer to Section 5 for the discussion and improvement.
Comment 3: Before resubmitting the paper, please do a proofreading carefully.
Answer 3: Thanks for this comment. We have done the proofreading for this revision. Besides, the original manuscript was polished by a native English speaker.
Round 2
Reviewer 2 Report
The modified parts reply my concerns in a proper manner. I think this revision is satisfactory.